# Streamlining *Bacillus* Strain Selection Against *Listeria monocytogenes* Using a Fluorescence-Based Infection Assay Integrated into a Multi-Tiered Validation Pipeline

**DOI:** 10.3390/antibiotics14080765

**Published:** 2025-07-29

**Authors:** Blanca Lorente-Torres, Pablo Castañera, Helena Á. Ferrero, Sergio Fernández-Martínez, Suleiman Adejoh Ocholi, Jesús Llano-Verdeja, Farzaneh Javadimarand, Yaiza Carnicero-Mayo, Amanda Herrero-González, Alba Puente-Sanz, Irene Sainz Machín, Isabel Karola Voigt, Silvia Guerrero Villanueva, Álvaro López García, Eva Martín Gómez, James C. Ogbonna, José M. Gonzalo-Orden, Jesús F. Aparicio, Luis M. Mateos, Álvaro Mourenza, Michal Letek

**Affiliations:** 1Departamento de Biología Molecular, Área de Microbiología, Universidad de León, 24071 León, Spain; blort@unileon.es (B.L.-T.); pcase@unileon.es (P.C.); halvf@unileon.es (H.Á.F.); sfernm14@estudiantes.unileon.es (S.F.-M.); jllav@unileon.es (J.L.-V.); fjavad00@estudiantes.unileon.es (F.J.); ycarm@unileon.es (Y.C.-M.); irenesanzm98@gmail.com (I.S.M.); isabel.karola.voigt@gmail.com (I.K.V.); guerrerovillanuevasilvia@gmail.com (S.G.V.); alopeg15@estudiantes.unileon.es (Á.L.G.); emartg10@estudiantes.unileon.es (E.M.G.); manolo@unileon.es (J.M.G.-O.); jesus.aparicio@unileon.es (J.F.A.); luis.mateos@unileon.es (L.M.M.); 2Department of Microbiology, University of Nigeria, University Road 1, Nsukka 410105, Nigeria; adejohsuleiman@yahoo.com (S.A.O.); james.ogbonna@unn.edu.ng (J.C.O.); 3Neural Therapies SL. Edif. Institutos de Investigación, Planta baja, Local B43. Campus de Vegazana s/n, 24071 León, Spain; aherrg03@estudiantes.unileon.es (A.H.-G.); apuens00@estudiantes.unileon.es (A.P.-S.); 4Instituto de Biología Molecular, Genómica y Proteómica (INBIOMIC), Universidad de León, 24071 León, Spain; 5Instituto de Desarrollo Ganadero y Sanidad Animal (INDEGSAL), Universidad de León, 24071 León, Spain

**Keywords:** *Listeria monocytogenes*, *Bacillus* spp., probiotic screening, fluorescence-based infection model, CFU validation, bacteriocins, antimicrobial pipeline, in vivo protection, host-pathogen interaction, microbial biocontrol

## Abstract

**Background/Objectives**: *Listeria monocytogenes* is a foodborne pathogen of major public health concern due to its ability to invade host cells and cause severe illness. This study aimed to develop and validate a multi-tiered screening pipeline to identify *Bacillus* strains with probiotic potential against *L. monocytogenes*. **Methods**: A total of 26 *Bacillus* isolates were screened for antimicrobial activity, gastrointestinal resilience, and host cell adhesion. A fluorescence-based infection assay using mCherry-expressing HCT 116 cells was used to assess cytoprotection against *L. monocytogenes* NCTC 7973. Eight strains significantly improved host cell viability and were validated by quantification of intracellular CFU. Two top candidates were tested in a murine model of listeriosis. The genome of the lead strain was sequenced to evaluate safety and biosynthetic potential. **Results**: *B. subtilis* CECT 8266 completely inhibited intracellular replication of *L. monocytogenes* in HCT 116 cells, reducing bacterial recovery to undetectable levels. In vivo, it decreased splenic bacterial burden by approximately 6-fold. Genomic analysis revealed eight bacteriocin biosynthetic clusters and silent antibiotic resistance genes within predicted genomic islands, as determined by CARD and Alien Hunter analysis. The strain also demonstrated bile and acid tolerance, as well as strong adhesion to epithelial cells. **Conclusions**: The proposed pipeline enables efficient identification of probiotic *Bacillus* strains with intracellular protective activity. *B. subtilis* CECT 8266 is a promising candidate for translational applications in food safety or health due to its efficacy, resilience, and safety profile.

## 1. Introduction

*Listeria monocytogenes* is a Gram-positive, facultative intracellular pathogen responsible for listeriosis, a severe foodborne disease with high mortality, particularly among immunocompromised individuals, pregnant women, and the elderly [1,2,3,4]. The ability of *L. monocytogenes* to replicate at refrigeration temperatures, tolerate high salt concentrations, and form biofilms underlies its persistence in food production environments and contributes to contamination of ready-to-eat products [5,6,7,8,9,10,11]. As a result, there is a growing interest in identifying bacterial strains or derived compounds that can actively prevent *L. monocytogenes* invasion and replication within host cells [12]. However, the lack of predictive, high-throughput functional assays has limited the efficiency with which such candidates can be identified and prioritised for translational applications.

Among candidate genera, *Bacillus* spp. have emerged as promising antagonists of *L. monocytogenes* due to their ability to produce diverse antimicrobial compounds, tolerate gastrointestinal conditions, and colonise mucosal surfaces [13,14,15,16,17,18]. Strains belonging to *Bacillus* spp. are particularly attractive for food and health-related applications, in part because of their Generally Recognised as Safe (GRAS) status and robustness under industrial and physiological stressors [19,20,21]. Numerous studies have reported bacteriocins and lipopeptides from *Bacillus* strains with potent anti-*Listeria* activity [10,22,23,24,25]. However, most of these findings are based on agar diffusion assays or bacteriocin purification protocols, which provide limited insight into functional efficacy within host-associated contexts. Crucially, the capacity of *Bacillus* strains to prevent host cell invasion by *L. monocytogenes* has rarely been assessed using standardised, predictive infection models. Recent studies have explored host cell-based screening methods to evaluate the efficacy of probiotic and biocontrol candidates against foodborne pathogens in vitro [13,26].

In this study, we developed and validated an integrated pipeline to identify *Bacillus* strains with protective activity against *L. monocytogenes*. Starting from a diverse panel of isolates, we applied sequential selection criteria including in vitro antimicrobial activity, survival under simulated gastrointestinal conditions, and host cell adhesion. A fluorescence-based infection model was then used as a rapid screening tool to identify strains that confer intracellular protection, with results corroborated by traditional CFU enumeration in a subset of candidates. The most promising strains were further tested in a murine challenge model, and genomic analysis was performed to assess bacteriocin potential and safety attributes. This stepwise approach streamlines the selection of probiotic *Bacillus* strains by ensuring functional efficacy, safety, and regulatory compliance of the candidates.

## 2. Results

### 2.1. Agar Well Diffusion Screening of Bacillus spp. Reveals Anti-Listeria Activity

The antimicrobial activity of 20 *Bacillus* spp. strains from the CECT collection (Appendix A) was evaluated using agar well diffusion assays with cell-free culture supernatants, thereby focusing on secreted bioactive compounds. Supernatants were tested against a panel of Gram-positive and Gram-negative pathogens, including *Staphylococcus aureus*, *Salmonella enterica* serovar Typhimurium, and *Listeria monocytogenes* (Figure 1 and Appendix A). Twelve strains showed inhibitory activity against at least one target organism, with nine exhibiting clear zones of inhibition against *L. monocytogenes* NCTC 7973 (Figure 1). To expand the strain repertoire, we screened eight African fermented foods, recognised as rich microbial reservoirs [27]. Four products—*Ogiri*, *Iru*, *Okpeye*, and *Ukpaka*—yielded 22 Gram-positive, spore-forming rod-shaped isolates. Among them, six strains, all isolated from *Ogiri* (OG) or *Okpeye* (OK), produced antimicrobial compounds effective against *L. monocytogenes* NCTC 7973 (Figure 1). These results allowed the initial selection of strains exhibiting strong anti-listerial potential for further testing in gastrointestinal-like conditions.

### 2.2. Tolerance to Gastrointestinal Stressors: Bile Salts and pH

Although *Bacillus* endospores are known for their resilience [28], the ability of vegetative cells to tolerate gastrointestinal stressors is a more relevant indicator of in situ activity. We therefore assessed the capacity of all *Bacillus* spp. strains to grow in the presence of bile salts (up to 0.5%) and at acidic pH values (as low as pH 3.5) to simulate small intestinal conditions. In adults, bile salt concentrations in the duodenum can reach ~10 mM (≈0.46% *w*/*v*) postprandially, while pH increases rapidly from ~3 in the stomach to ~7 in the upper small intestine, reflecting a major physicochemical transition relevant for probiotic survival [29].

As expected, bile salts inhibited vegetative growth to varying degrees. Notably, *B. subtilis* CECT 8266 displayed the highest tolerance among CECT-derived strains, maintaining measurable growth even at 0.5% bile salts (Figure 2). In contrast, several food-derived strains, including OK2 and OK3, were more susceptible to bile stress.

Acid tolerance was evaluated by monitoring bacterial growth at pH values ranging from 3.5 to 9.0. Most strains grew optimally between pH 5.0 and 7.0, with reduced proliferation observed at pH 9.0 and a marked decrease at pH 3.5 (Figure 3). Among the tested strains, *B. subtilis* CECT 40 showed the broadest pH tolerance, although several other isolates—including CECT 8266—also sustained growth under acidic conditions.

Thirteen strains that exhibited both anti-*Listeria* activity and resilience to pH and bile salt stress were selected for downstream functional evaluation using host cell-based infection assays. All food-derived isolates chosen for further analysis were taxonomically identified at this stage as *B. subtilis* based on 16S rDNA and *tuf* gene sequencing (Appendix A).

### 2.3. A Fluorescence-Based Infection Assay Determines Host Cell Protection Against L. monocytogenes

To streamline the identification of protective strains, a fluorescence-based infection model was used to screen preselected candidates for their ability to protect host cells from *L. monocytogenes*–induced cytotoxicity. We applied a high-throughput, fluorescence-based assay using HCT 116 human intestinal epithelial cells transduced for constitutive expression of the red fluorescent protein mCherry. This system enables the direct quantification of host cell viability following bacterial challenge, independent of host metabolism, which can be compromised during infection and lead to unreliable results in conventional assays based on the reduction of MTT or lactate dehydrogenase release [30,31].

We first validated the assay by correlating mCherry fluorescence intensity with MTT-derived viability measurements across varying cell densities. A strong linear correlation was observed (Figure 4; correlation coefficient = 0.92), confirming that mCherry fluorescence is a reliable proxy for cell viability under our experimental conditions.

Next, we applied the assay to assess whether pre-exposure to *Bacillus* strains conferred protection to mCherry-HCT 116 cells infected with *L. monocytogenes*. Thirteen strains were tested, and mCherry fluorescence was measured at 20 h post-infection as an indicator of host cell survival. Eight strains significantly improved cell viability compared to infected, untreated controls (C-LMO), indicating effective protection against pathogen-induced cytotoxicity (Figure 5). Notably, five of these strains—despite showing strong antimicrobial activity in agar diffusion assays—failed to protect host cells, suggesting that in vitro antimicrobial activity alone is not a sufficient predictor of functional efficacy in infection models.

To validate the performance of the fluorometric assay, we quantified intracellular bacteria via CFU enumeration for the top eight candidates. *B. subtilis* CECT 8266 completely inhibited intracellular bacterial replication at both 4- and 20-h post-infection (Figure 6), indicating potent blockade of pathogen internalisation or early intracellular survival. *B. subtilis* OK3 also reduced intracellular burden, but the effect was less pronounced.

Together, these results demonstrate that the fluorescence-based infection assay provides a reliable and high-throughput platform to assess the ability of *Bacillus* strains to protect host epithelial cells from *L. monocytogenes*-induced cytotoxicity. The observed differences between strains with comparable in vitro antimicrobial activity highlight the importance of functional screening in host-relevant models.

### 2.4. In Vivo Validation of Protective Strains in a Murine Model of Listeriosis

To evaluate whether the protective effects observed in vitro translated into host protection in vivo, we selected two top-performing strains—*B. subtilis* CECT 8266 and OK3—for testing in a murine model of *L. monocytogenes* infection. These strains were prioritised based on their consistent performance across multiple in vitro assays, as they exhibited strong inhibition zones against *L. monocytogenes*, demonstrated tolerance to gastrointestinal stressors, and provided substantial protection to HCT 116 enterocytes in the fluorescence-based infection assay. In addition, both *B. subtilis* CECT 8266 and OK3 inhibited multiple *L. monocytogenes* isolates in agar diffusion assays (Appendix A), suggesting a broad inhibitory spectrum that may further enhance their translational relevance.

Therefore, BALB/c mice were orally administered either *B. subtilis* CECT 8266, OK3, or PBS for ten consecutive days before challenge with *L. monocytogenes* NCTC 7973. Mice were monitored for weight loss and signs of disease, and were sacrificed 48 h post-infection for quantification of the bacterial burden in the spleen. Survival was also monitored throughout the experiment. One mouse in the infected, untreated control group died before the experimental endpoint was reached. All other animals, including those treated with *B. subtilis* strains, survived until necropsy. Both *Bacillus*-treated groups tolerated the strains well, with no weight loss or signs of discomfort observed in uninfected animals (Figure 7A), confirming the safety of these strains under the tested conditions.

In infected animals, weight loss was significantly reduced in the group treated with *B. subtilis* CECT 8266 compared to the infected, untreated controls (Figure 7A). Moreover, spleen homogenates from this group revealed a ~6-fold reduction in *L. monocytogenes* CFU counts relative to both the untreated group and the OK3-treated group (Figure 7B). In contrast, mice treated with *B. subtilis* OK3 exhibited no significant reduction in splenic bacterial burden, despite the moderate protection observed in vitro. These findings suggest that *B. subtilis* CECT 8266 confers a biologically meaningful protection against *L. monocytogenes* infection in vivo, as evidenced by improved survival, reduced weight loss, and lower bacterial burden.

### 2.5. Host Cell Adhesion and Growth Dynamics

To further explore traits that may contribute to host protection, we assessed bacterial adhesion to HCT 116 cells and compared growth kinetics across selected *Bacillus* strains. Adhesion was quantified by co-incubating bacteria with HCT 116 monolayers, followed by extensive washing and CFU enumeration. Both *B. subtilis* CECT 8266 and OK3 exhibited significantly higher adhesion to host cells compared to the type strain *B. subtilis* CECT 39T (Figure 8A), suggesting enhanced capacity for mucosal colonisation.

We also monitored bacterial growth in rich media by measuring absorbance at 600 nm (Abs_600_) over time. Growth kinetics were analysed by plotting ln(Abs_600_) versus time during the exponential phase and applying linear regression to estimate the specific growth rate (μ, h^−1^). The resulting slopes indicated similar exponential growth rates across the three strains: *B. subtilis* CECT 39T (μ = 1.295 ± 0.051), *B. subtilis* CECT 8266 (μ = 1.266 ± 0.047), and *B. subtilis* OK3 (μ = 1.292 ± 0.030), with no statistically significant differences observed between them.

Despite these comparable growth rates, final biomass accumulation differed. *B. subtilis* OK3 reached the highest absorbance during stationary phase (1.4 ± 0.1), followed by *B. subtilis* CECT 8266 (1.2 ± 0.0), and *B. subtilis* CECT 39T (0.9 ± 0.2). However, these in vitro parameters did not correlate with in vivo performance. Only *B. subtilis* CECT 8266 conferred protection in the murine infection model, despite similar growth rates across all strains and higher biomass in OK3. These results indicate that growth kinetics and biomass accumulation alone do not explain the protective effect of *B. subtilis* CECT 8266, highlighting the importance of strain-specific factors in its probiotic potential.

Previous studies have suggested that strong epithelial adhesion may support the competitive exclusion of *L. monocytogenes* in the gut [26]. However, in our study, adhesion capacity and growth rate were not reliable predictors of protective efficacy. For example, *B. subtilis* OK3 exhibited robust epithelial adhesion and rapid growth but failed to confer full protection in both in vitro and in vivo infection models.

### 2.6. Evidence of Bacteriocin-Mediated Antimicrobial Activity in B. subtilis CECT 8266

Numerous studies have suggested that the anti-listerial activity of *Bacillus* strains is mediated by the production of bacteriocins [25,32,33,34,35,36,37,38,39,40]. To assess whether extracellular bacteriocins contribute to the antimicrobial activity of *B. subtilis* CECT 8266, we compared its secreted protein fractions with those of *B. subtilis* CECT 561, a strain lacking detectable activity against *L. monocytogenes* in agar diffusion assays and thus used as a negative control (Figure 1).

Both strains were processed in parallel following established protocols for bacteriocin purification [41,42,43]. Ammonium sulphate precipitation was selected as the initial step for recovering bacteriocin activity due to its mild, tunable conditions, which allowed for the successful recovery of bioactive peptides from *B. subtilis* CECT 8266. In contrast, other methods, such as trichloroacetic acid or ethanol precipitation, did not yield active fractions under our experimental conditions [38,44,45]. It is important to note that this approach enriches the proteinaceous fraction of the culture supernatant, selectively concentrating bacteriocins while excluding low-molecular-weight antimicrobial compounds that remain soluble [38,44].

Ammonium sulphate precipitation at 60% and 80% saturation successfully concentrated active compounds from *B. subtilis* CECT 8266 supernatants, yielding inhibition halos of 26 mm and 27 mm, respectively (Figure 9). In contrast, no antimicrobial activity was detected in equivalent fractions from *B. subtilis* CECT 561.

Attempts to recover active fractions using trichloroacetic acid or ethanol precipitation were unsuccessful for both strains, likely due to protein denaturation or poor solubility. These results indicate that the observed anti-listerial activity of *B. subtilis* CECT 8266 is due to secreted proteinaceous compounds—consistent with bacteriocins—that are stable under mild ammonium sulphate precipitation conditions but are inactivated by harsher extraction methods.

### 2.7. Genomic Analysis of B. subtilis CECT 8266 Reveals Bacteriocin-Encoding Genes

To better understand the molecular basis of the pronounced anti-listerial activity exhibited by *B. subtilis* CECT 8266 both in vitro and in vivo, we sequenced its genome (Appendix A). The resulting assembly spans 4,163,299 base pairs and includes 4145 predicted coding sequences.

AntiSMASH and BAGEL5 analyses revealed a diverse array of biosynthetic gene clusters (BGCs) associated with secondary metabolite production (Appendix A). These include non-ribosomal peptide synthetase (NRPS) clusters for surfactin, fengycin, and bacillibactin biosynthesis—compounds with documented antimicrobial, antifungal, and iron-scavenging functions [46,47,48]. Additional BGCs encode bacillaene, subtilosin A, pulcherriminic acid, bacilysin, and terpenes, many of which have demonstrated antibacterial activity [17,49,50,51]. Several terpene BGCs were also detected, suggesting the potential for volatile bioactive compound production [52]. Furthermore, a Type III polyketide synthase (T3PKS) gene cluster of unknown function was present.

All of these genomic features are part of the *Bacillus* pangenome, which suggests that the potent anti-listerial activity of *B. subtilis* CECT 8266 is not due to the presence of novel or unique bacteriocin genes. Instead, it likely results from the combined action of multiple well-characterised and conserved anti-listerial compounds—such as subtilosin A, surfactin, and fengycin [35,36,38]—whose synergistic interactions may underlie the robust phenotype observed.

### 2.8. Antibiotic Resistance

Antibiotic resistance must be absent for *Bacillus* strains to be considered safe candidates for microbial-based therapies or supplements, in line with the safety criteria established by the European Food Safety Authority (EFSA) and recent evaluations of probiotic suitability [19,53]. Therefore, we evaluated the antibiotic susceptibility of *B. subtilis* CECT 8266 by determining its Minimum Inhibitory Concentrations (MIC) values against a panel of EFSA-recommended antibiotics (Appendix A) [53]. No resistance was observed, including to aminoglycosides and glycopeptides, two critically important classes of antibiotics. In contrast, the type strain *B. subtilis* CECT 39T exhibited resistance to streptomycin and clindamycin. Notably, *B. subtilis* OK3 exhibited resistance to all tested antibiotics except chloramphenicol, likely reflecting the historical lack of antibiotic regulation in Nigeria, which has only recently begun to be addressed [54].

To complement this phenotypic assessment, we screened the genome of *B. subtilis* CECT 8266 using the Comprehensive Antibiotic Resistance Database (CARD). Several genes commonly associated with antimicrobial resistance were identified (Appendix A). However, these genes represent either intrinsic features widely conserved in *Bacillus* spp. (e.g., efflux pumps or regulatory proteins), or distant homologs of acquired resistance determinants not known to be functional in this genus. Importantly, although a subset of these genes (*vanT*, *vanW*, *BcI*, and *aadK*) were located within regions predicted to be horizontally acquired, no corresponding phenotypic resistance to vancomycin or streptomycin was detected in MIC testing. These findings indicate that these genomic features do not confer clinically relevant resistance in this context.

Alien Hunter analysis detected several horizontally acquired genomic islands, characterized by altered GC content and absent from the reference genomes (Appendix A). One of these regions, spanning positions 4,004,525 to 4,009,686, encodes several hypothetical proteins and a bacitracin transport ATP-binding protein (BcrA3). This is one of three *bcrA* homologs in the genome, all of which are implicated in bacitracin resistance.

Together, the combined genomic and phenotypic data confirm that *B. subtilis* CECT 8266 does not harbour acquired antimicrobial resistance traits of safety concern. These results meet EFSA criteria, which prioritise the absence of transferable resistance genes and phenotypic resistance to clinically relevant antibiotics.

## 3. Discussion

Our results demonstrate that *Bacillus* strains exhibit pronounced specificity against *L. monocytogenes* (Appendix A), likely reflecting an evolutionary adaptation driven by ecological niche competition [55]. *B. subtilis* and *L. monocytogenes* frequently co-occur in soil, silage, fermented foods, and food-processing environments, where persistent cohabitation may have selected for *B. subtilis* strains capable of producing targeted antimicrobials such as bacteriocins and lipopeptides [56,57,58,59]. This selective pressure may explain why strains like *B. subtilis* CECT 8266 display strong and specific anti-listerial activity. Notably, *B. subtilis* can allocate up to 5% of its genome to the synthesis of secondary metabolites with antimicrobial function [60].

Interestingly, all food-derived isolates that progressed through our screening were identified as *B. subtilis* based on 16S rRNA and *tuf* gene sequencing. This aligns with its known dominance in traditional fermentations and its ability to produce antimicrobials in situ [61]. Such prevalence suggests fermented foods are a valuable reservoir of anti-*Listeria Bacillus* strains. While this specificity is advantageous for food safety, the limited activity against Gram-negative bacteria implies minimal disruption of the gut microbiota, which is desirable in the context of targeted probiotic applications [62].

Our stepwise screening pipeline revealed that anti-listerial activity alone was insufficient to qualify a strain as a probiotic candidate. Many isolates failed to tolerate low pH or bile salts—key barriers to gastrointestinal survival—leading to a high attrition rate, in line with previous probiotic discovery efforts [63]. Only about half of the strains met all basic functional criteria, underscoring the importance of combining antimicrobial testing with resilience assays early in the selection process.

Antibiotic susceptibility is another critical safety parameter. According to EFSA guidelines, probiotic strains must not harbour acquired resistance genes [53]. While *B. subtilis* OK3 showed promising antimicrobial effects, it exhibited multidrug resistance, disqualifying it as a safe candidate. In contrast, *B. subtilis* CECT 8266 was sensitive to all tested antibiotics. This clean profile, together with acid and bile tolerance and epithelial adhesion, made *B. subtilis* CECT 8266 the only isolate (~4% of our initial pool) to meet all probiotic selection criteria—consistent with previous reports indicating that true candidates emerge at low frequency [63].

Remarkably, in vitro inhibition of *L. monocytogenes* did not always translate to protection in host cell assays. Approximately one-third of the strains with strong agar diffusion activity failed to prevent infection in the mCherry-based fluorescence assay. This limitation is well-documented: pathogen inhibition in bacteriological assays does not necessarily reflect protection of host cells or modulation of infection outcomes [64,65]. Functional screening in physiologically relevant models is therefore essential. This approach not only refines selection but also reduces the number of strains requiring validation in animal models, aligning with the 3Rs principle (Replacement, Reduction, and Refinement) [66].

In the murine model of listeriosis, only *B. subtilis* CECT 8266 reduced splenic bacterial burden and mitigated weight loss. Full eradication is rare in oral *Listeria* challenge models [67]. Therefore, we considered a ~6-fold reduction in spleen CFUs, coupled with improved clinical signs and 100% survival, a biologically meaningful effect. Variability in organ colonisation and technical limitations in tissue homogenization may have contributed to noise in pathogen recovery, but the spleen data and weight monitoring support the protective capacity of *B. subtilis* CECT 8266. Notably, our protocol employed a prophylactic approach, with *Bacillus* administered before infection. Future studies should investigate the continuous administration of this approach to optimise efficacy.

Whole-genome sequencing of *B. subtilis* CECT 8266 revealed eight biosynthetic gene clusters linked to anti-listerial activity, including subtilosin A, surfactin, and fengycin [35,36,38]. These compounds likely act synergistically to produce the observed inhibition. While our extraction methods favoured proteinaceous antimicrobials such as bacteriocins, we may have missed hydrophobic metabolites like polyketides or organic acids. Further transcriptomic or metabolomic profiling will be necessary to fully characterise the active antimicrobial components.

Crucially, genome mining using CARD and Alien Hunter, combined with MIC determination, confirmed the absence of functional resistance determinants. Although genes such as *vanT*, *vanW* or *aadK*, were detected, none conferred phenotypic resistance to vancomycin, streptomycin or kanamycin—suggesting they are either non-functional or not expressed, a finding consistent with reports in other bacteria [68]. This supports the regulatory safety profile of *B. subtilis* CECT 8266.

As a spore-former, *B. subtilis* CECT 8266 is suitable for oral administration and formulation as a stable supplement or food additive [69,70,71]. Given its genetic stability, the strain could be further improved through synthetic biology—for instance, by overexpressing antimicrobial biosynthetic genes or deleting latent resistance loci—to enhance both efficacy and safety.

## 4. Materials and Methods

### 4.1. Bacterial Strains and Culture Conditions

We screened twenty *Bacillus* spp. strains from the *Colección Española de Cultivos Tipo* (CECT) for attributes linked to gut colonisation and antimicrobial activity; genome data are publicly available for four of them (Appendix A). The *L. monocytogenes* strains used in this study included NCTC 7973 (reference strain used for infection models), ATCC 7644, CECT 911 (ATCC 19112; serotype 1/2c, cerebrospinal fluid isolate), CECT 940 (ATCC 19117; sheep isolate), CECT 4031 (ATCC 15313; rabbit isolate), and ST4C (serotype 4c; poultry isolate). Strain information and origin are detailed in Appendix A.

All these strains were cultured on Trypticase Soy Agar or Broth (TSA or TSB; Biokar Diagnostics, Solabia Group, Paris La Défense, France) and incubated at 37 °C, with shaking at 200 rpm for liquid cultures. All remaining bacterial strains used here were cultured on Brain Heart Infusion (BHI) Agar or Broth (Condalab, Laboratorios Conda S.A., Torrejón de Ardoz, Madrid, Spain), also at 37 °C and shaking at 200 rpm for liquid cultures, as previously reported [72].

We also expanded our collection by including additional strains sourced from eight traditional Nigerian fermented foods and condiments: *Ogiri*, *Iru*, *Okpeye*, *Ukpaka*, *Garri*, *Fufu*, *Owo*, and *Fura* (Appendix A). For this, one gram of each solid sample was mixed with 9 mL of distilled water and thoroughly shaken. The mixtures were then heated in a water bath at 80 °C for 15 min to eliminate vegetative (non-spore-forming) bacteria. After heat treatment, 100 μL of each suspension was spread onto TSA plates and incubated at 37 °C to promote bacterial growth. Colonies with distinct morphologies were carefully selected and subcultured for further analysis. The identity of *Bacillus* species among the isolates was confirmed by Gram staining, followed by sequencing of the 16S rDNA and *tuf* genes (see Section 4.10).

Growth curves of *B. subtilis* strains were obtained by culturing the bacteria in BHI medium using 96-well U-bottom plates. Each well was inoculated with a 1:1000 dilution of an overnight culture adjusted to an absorbance measured at a wavelength of 600 nm (Abs_600_) of 1. Plates were incubated at 37 °C with continuous shaking in a Victor Nivo plate reader (Revvity, Waltham, MA, USA). Abs_600_ was recorded hourly over 24 h to monitor bacterial growth.

Bacterial growth was monitored by measuring absorbance at 600 nm (Abs_600_) at regular time intervals using a spectrophotometer. To determine the specific growth rate (μ, in h^−1^), a semi-logarithmic approach was applied. Absorbance values were transformed to their natural logarithm (ln Abs_600_), and the exponential growth phase for each strain was visually identified. A simple linear regression was applied to the ln-transformed Abs_600_ values as a function of time (t, in hours), using the equation: ln(Abs_600_) = μ × t + b, where μ is the slope of the line (specific growth rate) and b is the intercept. The exponential growth phase was defined as follows: for *B. subtilis* CECT 39T and *B. subtilis* CECT 8266, it was between 5 and 8 h; for *B. subtilis* OK3, it was between 2 and 9 h.

### 4.2. Agar-Well Diffusion Tests and Determination of the Minimum Inhibitory Concentrations

The antimicrobial activity of the selected *Bacillus* spp. strains was evaluated against various bacteria using the agar-well diffusion method. This assay was conducted according to the guidelines provided by the Clinical and Laboratory Standards Institute (CLSI) for antimicrobial susceptibility testing [73], with some modifications. The day before the assay, *Bacillus* spp. strains were cultured in TSB. On the day of the assay, 2 mL of each culture were collected, centrifuged in Eppendorf tubes at maximum speed for 10 min, and the supernatants were filtered through 0.45 µm pore filters using 5 mL syringes, except for *B. subtilis* CECT 39T and CECT 371, which had to be filtered with 0.22 µm pore filters to prevent bacterial growth in the borders of the wells. The filtrates were transferred to sterile Eppendorf tubes for immediate use. Although samples could be stored at 4 °C for up to 24 h without significant loss of activity, prolonged storage resulted in a marked reduction of antimicrobial efficacy. Notably, freezing the cell-free supernatants completely abolished their antimicrobial activity.

A range of bacterial strains was prepared for the assay by embedding them as test strains in agar plates. Test strains were cultured in TSB the day before and harvested at an Abs_600_ in the range of 0.8 to 1.2. The final volume added to the bioassay medium was adjusted to an Abs_600_ of 0.03. Then, TSA was melted and cooled to around 45 °C and 2 mL of the TSB medium containing the test strain was added to 100 mL of TSA agar, mixed thoroughly, and poured into four 100 mm Petri dishes. Once solidified, equidistant wells were made in the agar using the mouth of a sterile Pasteur pipette, and the plugs were removed with sterile toothpicks. Each well was filled with 35 µL of the *Bacillus* spp. cell-free supernatants, and the plates were incubated at 37 °C for 24 h. The plates were then examined for the presence of inhibition halos, and their diameters were measured with a calliper.

The MIC of a selected panel of EFSA-recommended antibiotics was determined following the CLSI guidelines [73]. Briefly, *Bacillus* spp. strains were cultured in Mueller-Hinton broth to the exponential growth phase (Abs_600_ = 1), then diluted to a final concentration of 2 × 10^5^ cells per 100 µL in each well of 96-well microtiter plates. Serial dilutions of the antibiotics were prepared in triplicate, and the plates were incubated at 37 °C for 16 h. Each experiment included a negative control containing only Mueller-Hinton broth.

### 4.3. Biliary Salts and pH Tests

Vegetative growth of *Bacillus* spp. strains was assessed under varying pH conditions (3.5, 5.0, 7.0, and 9.0) and bile salt concentrations (0–0.5%). For each condition, 5 µL of overnight cultures were inoculated into 96-well plates containing 95 µL of TSB adjusted accordingly. Cultures were incubated at 37 °C with shaking for 24 h, and growth was quantified by measuring absorbance at 600 nm using a Victor Nivo plate reader. Values were normalised against blank wells containing only media and compared to control conditions (pH 7.0 or TSB without bile salts).

### 4.4. Cellular Lines and Culture Conditions

Human epithelial colon HCT 116 cells (CCL-247, American Type Culture Collection, Manassas, VA, USA) were cultured in 100 mm cell culture plates with Dulbecco’s Modified Eagle’s Medium (DMEM; Gibco, Thermo Fisher Scientific, Waltham, MA, USA) supplemented with pyruvate, glucose, glutamine, 10% heat-inactivated foetal bovine serum (FBS; Gibco), and 5% penicillin-streptomycin solution (Corning, NY, USA). HCT 116 cells expressing constitutively mCherry were generated by transduction with the p12-MMP-mCherry vector as described previously [74]. To ensure selective growth, hygromycin (Sigma-Aldrich, MilliporeSigma, St. Louis, MO, USA) was added to the medium to eliminate non-transduced cells. The cells were maintained at 37 °C in a 5% CO_2_ atmosphere. The expression of mCherry was confirmed by measuring fluorescence in a Victor Nivo plate reader with an excitation filter of 580 nm and an emission filter of 625 nm [75].

### 4.5. Cell Viability Assay

HCT 116 cells expressing mCherry were seeded at varying densities in black-walled, clear-bottom 96-well plates containing complete DMEM. The plate was incubated at 37 °C with 5% CO_2_ for 24 h and then washed twice with phosphate-buffered saline (PBS; Gibco, Thermo Fisher Scientific, Waltham, MA, USA). The mCherry fluorescence was measured as described above. Then, PBS was replaced with 100 µL of DMEM containing antibiotics, and 10 µL of tetrazolium dye 3-(4,5-dimethylthiazol-2-yl)-2,5-diphenyltetrazolium bromide (MTT) solution (Alfa Aesar, Thermo Fisher Scientific, Ward Hill, MA, USA) was added to each well. The plate was incubated for 4 h at 37 °C with 5% CO_2_. After incubation, 100 µL of isopropanol (Fisher BioReagents, Thermo Fisher Scientific, Pittsburgh, PA, USA) with 0.04 N HCl (Fisher Chemical, Thermo Fisher Scientific, Waltham, MA, USA) was added to each well, and the plate was incubated for an additional hour. Absorbance was then measured at 570 nm with a reference wavelength of 630 nm using a Victor Nivo plate reader.

### 4.6. Infection Assay

HCT 116 cells were seeded in 96-well plates (8 × 10^4^ cells per well) with complete DMEM medium (without antibiotics) and incubated O/N at 37 °C with 5% CO_2_. On the day of the assay, the medium was replaced with 50 µL of antibiotic-free media containing *Bacillus* spp. strains, adjusted to a final concentration of 8 × 10^5^ CFUs per well. The plate was centrifuged at 800× *g* for 5 min, then incubated at 37 °C with 5% CO_2_ for 2 h. Following this incubation, the cells were washed with PBS and subsequently infected with *L. monocytogenes* at a multiplicity of infection (MOI) of 0.1. For this, 150 µL of antibiotic-free media containing the adjusted concentration of *L. monocytogenes* was added to each well. The plate was centrifuged again and incubated under the same conditions. After the incubation period, the medium was replaced with complete DMEM containing gentamicin sulphate (100 µg/mL; MP Biomedicals, Santa Ana, CA, USA) to eliminate extracellular bacteria. The fluorescence of viable mCherry-expressing cells was measured as described above at 20 h post-infection.

To determine the number of intracellular CFUs of *L. monocytogenes*, HCT 116 cells were seeded in a 24-well plate at a density of 2.4 × 10^5^ cells per well and incubated overnight at 37 °C with 5% CO_2_. On the day of the assay, the medium was replaced with 500 µL of media containing *Bacillus* spp. strains, adjusted to a concentration of 2.4 × 10^6^ CFUs per well. The plate was centrifuged at 800× *g* for 5 min and then incubated at 37 °C with 5% CO_2_ for 2 h. Following incubation, the cells were washed with PBS, and 500 µL of antibiotic-free DMEM containing *L. monocytogenes*, adjusted to an MOI of 0.1, was added to each well. The plate was centrifuged and incubated as previously described. After incubation with *Bacillus* strains and *L. monocytogenes*, the medium was replaced with complete DMEM containing gentamicin (100 µg/mL) to kill any extracellular bacteria. At 4 and 20 h post-infection, the cells were washed with PBS and lysed with 100 µL of 0.1% Triton X-100 (Sigma-Aldrich, MilliporeSigma, St. Louis, MO, USA) for 3 min at room temperature. After lysis, 900 µL of PBS was added to each well. The lysates were then serially diluted and plated on BHI agar for CFU counting.

### 4.7. Adhesion Assay

HTC 116 cells were seeded in 24-well plates at a density of 1.5 × 10^5^ cells per well in complete DMEM. On the day of the experiment, *Bacillus* spp. strains, previously cultured under standard conditions, were washed and resuspended in antibiotic-free DMEM. Bacterial suspensions were adjusted to an OD_600_ of 1.0. Cell monolayers were gently washed with PBS before adding 1 mL of a 1:10 dilution of the OD-normalised bacterial suspension (~1 × 10^7^ CFU/well; MOI ~10). Untreated wells served as negative controls. After one hour of incubation at 37 °C, the medium was aspirated, and wells were washed three times with PBS. Cells were lysed with 0.1% Triton X-100, and lysates were resuspended in 1 mL of PBS. Serial dilutions were plated on TSA to determine the number of adherent *Bacillus* spp. by CFU enumeration.

### 4.8. Evaluation of Protective Effects of Bacillus Strains in a Mouse Model of L. monocytogenes Infection

Forty-eight female BALB/c mice, 6–8 weeks old, were obtained from Charles River (France). Mice were housed in Type III cages in a dedicated biosecurity level 2 room. Cages were environmentally enriched with paper and cardboard, and food and water were provided *ad libitum*. The room was maintained under a 12-h light/dark cycle, with a relative humidity of 50–60% and a temperature of 20–25 °C. Mice were allowed to acclimatize for five days before the experiment and were randomly assigned to six groups, with eight mice per group. The experimental groups consisted of a PBS control, two groups administered *B. subtilis* CECT 8266 or *B. subtilis* OK3, and three matching groups that were subsequently challenged with *L. monocytogenes* NCTC 7973.

*B. subtilis* strains CECT 8266 and OK3 were cultured in TSB for 24 h. Cells were harvested by centrifugation, washed three times with PBS (pH 7.4), and resuspended to a final concentration of 5 × 10^11^ CFU/mL. A 100 µL suspension containing 5 × 10^10^ CFUs was administered orally by gavage once daily for 10 days to the designated groups, while the control groups received 100 µL of PBS.

*L. monocytogenes* NCTC 7973 was cultured in BHI for 12 h and prepared to a concentration of 5 × 10^10^ CFU/mL for BALB/c mice. On day 16, after a 5-h fasting period, 100 µL of a *L. monocytogenes* suspension containing 5 × 10^9^ CFUs was administered orally via gavage to the infected groups (namely control, CECT 8266, and OK3). Uninfected control groups received 100 µL of PBS.

Mice were monitored daily for weight loss, physical appearance, and gastrointestinal disturbances. Humane endpoints were established, including a weight loss exceeding 20%, signs of severe dehydration, or persistent diarrhoea lasting more than 48 h. All mice were sacrificed 48 h post-infection (day 18) using cervical dislocation. Following euthanasia, aseptic necropsies were performed. Spleens were collected and homogenised in 9 mL of PBS supplemented with 0.1% Triton X-100 (Sigma-Aldrich, MilliporeSigma, St. Louis, MO, USA). The number of CFUs in each organ was determined using Modified Oxford Agar plates, a selective medium for *L. monocytogenes* (Condalab, Laboratorios Conda S.A., Torrejón de Ardoz, Madrid, Spain) that was supplemented with an Oxford *Listeria* Selective Supplement (Condalab, Laboratorios Conda S.A., Torrejón de Ardoz, Madrid, Spain).

### 4.9. Purification of Bacteriocins

Purification of bacteriocins was performed following a previously described protocol using ammonium sulphate [41,42,43], with some modifications. *Bacillus* strains were cultured in 100 mL of TSB at 37 °C for 16 h with shaking. Cells were removed from the medium by centrifugation at 8210× *g* for 5 min. The cell-free supernatant was filtered through a 0.22 µm filter and subjected to protein precipitation using different agents: trichloroacetic acid (TCA), ethanol (EtOH), and ammonium sulphate. For TCA precipitation, either 10% or 6% (*w*/*v*) TCA was added to the supernatant and incubated at 4 °C for 2 h. For ethanol precipitation, two volumes of EtOH were added to one volume of supernatant, and the mixture was incubated at 4 °C overnight. For ammonium sulphate precipitation, 60% or 80% saturation was used, followed by incubation at 4 °C overnight. In all cases, the resulting pellets were washed twice with acetone and air-dried. The pellets were then resuspended in distilled water and dialysed against 4 L of deionised water overnight. The dialysed proteins were subsequently tested for antimicrobial activity, as described in Section 4.2.

### 4.10. DNA Sequencing, Annotation and Analysis

Pure cultures of *B. subtilis* CECT 8266 were grown in TSB, and harvested cells were resuspended in DNA/RNA Shield (Zymo Research, Irvine, CA, USA) and processed at MicrobesNG (Birmingham, UK). Cell suspensions (5 μL) were lysed with 120 μL of TE buffer containing lysozyme (MP Biomedicals, Irvine, CA, USA), metapolyzyme (Sigma-Aldrich, MilliporeSigma, St. Louis, MO, USA), and RNase A (ITW Reagents, Castellar del Vallès, Barcelona, Spain), followed by incubation at 37 °C for 25 min. Subsequently, Proteinase K (VWR Chemicals, Avantor, Radnor, PA, USA) was added to a final concentration of 0.1 mg/mL, and SDS (Sigma-Aldrich, MilliporeSigma, St. Louis, MO, USA) was added to a final concentration of 0.5% (*v*/*v*). This mixture was incubated at 65 °C for 5 min. Genomic DNA was purified using an equal volume of SPRI beads and resuspended in EB buffer (10 mM Tris-HCl, pH 8.0). Extracted DNA was quantified using the Quant-iT dsDNA HS assay (Thermo Fisher Scientific, Waltham, MA, USA) in an Eppendorf AF2200 plate reader (Eppendorf, Hamburg, Germany).

We then employed a hybrid sequencing approach, combining Illumina and Oxford Nanopore Technologies (ONT, Oxford, UK) sequencing methods. Illumina sequencing was performed on an Illumina NovaSeq 6000 platform (Illumina, San Diego, CA, USA) using a 250 bp paired-end protocol. ONT sequencing was conducted on a GridION platform. DNA libraries for Illumina sequencing were prepared using the Nextera XT Library Prep Kit (Illumina, San Diego, CA, USA), following the manufacturer’s protocol with modifications: the input DNA amount was doubled, and the PCR elongation time was extended to 45 s. DNA quantification and library preparation were automated using a Hamilton Microlab STAR system (Hamilton Bonaduz AG, Bonaduz, Switzerland). For ONT sequencing, high-molecular-weight DNA (200–400 ng) was used to prepare DNA libraries with the SQK-RBK114.96 kit (ONT, Oxford, UK). Barcoded samples were pooled into a single sequencing library and loaded onto a FLO-MIN114 (R.10.4.1) flow cell for sequencing on a GridION platform (ONT, Oxford, UK). Raw sequencing reads were adapter-trimmed using Trimmomatic version 0.30 with a sliding window quality cutoff of Q15 [76]. Hybrid de novo assembly of Illumina and ONT reads was performed using Unicycler version 0.4.0 [77].

The resulting contigs were annotated using Prokka version 1.11 [78]. Functional annotation was refined using InterProScan (v5.53) [79]. Functional pathway analysis was carried out using the KEGG Automatic Annotation Server (KAAS) to identify metabolic and signalling pathways [80]. Signal peptides were identified using SignalP (v5.0) [81]. Transmembrane domains were predicted using TMHMM (v2.0) [82]. Secondary metabolite biosynthetic gene clusters were predicted using antiSMASH (v6.0) [83]. Antimicrobial resistance genes were identified using the Comprehensive Antibiotic Resistance Database (CARD) (v3.2.0) [84]. Horizontal gene transfer regions were identified using Alien Hunter [85], and potential bacteriocins were predicted with Bagel 5 [86]. The complete genome sequence has been deposited in the NCBI database under BioSample accession number SAMN44006356.

All *Bacillus* isolates selected from fermented foods were taxonomically identified by sequencing both the 16S rDNA and *tuf* genes. For 16S rDNA, the universal primer pair 8FPL (5′-GCGGATCCGCGGCCGCTGCAGAGTTTGATCCTGGCTCAG-3′) and 806R (5′-GCGGATCCGCGGCCGCGGACTACCAGGGTATCTAAT-3′) was used [87]. For *tuf* gene amplification, the primer pair tuf2-F (5′-AVGGHTCTGCHYTDAAAGC-3′) and tuf2-R (5′-GTDAYRTCHGWWGTACGGA-3′) was employed, as described previously [88]. For Sanger sequencing, total DNA was extracted using a boiling method [87]. In summary, a loopful of a colony (10 µL) grown on TSB was suspended in 100 µL of water, boiled for 10 min, and then centrifuged for 2 min. The supernatant obtained was utilised for further analysis. Then, 16S rDNA genes were amplified into a 0.8 kb fragment using an Applied Biosystems Veriti Pro Thermal Cycler (Thermo Fisher Scientific, Foster City, CA, USA) and standard primers 8FPL/806R [87]. In addition, we also sequenced a 500 bp amplicon from *tuf* gene using primer pairs tuf2-F/tuf2-R [88]. The resulting gene fragments were directly used as templates for DNA sequencing, which was carried out using a 3500xL Genetic Analyser (Applied Biosystems, Thermo Fisher Scientific, Foster City, CA, USA). Similarity searches were conducted on public servers, provided by the National Centre for Biotechnology Information (Bethesda, MD, USA). Sequences were aligned using MAFFT (v7) with default settings for nucleotides [89]. Phylogenetic analysis was performed using the Neighbour-Joining method based on the Jukes-Cantor model for nucleotide substitution. Bootstrap resampling was carried out with 1000 replicates to assess the robustness of the tree topology. The resulting trees were visualised using Interactive Tree Of Life (iTOL) [90], and annotated manually for clarity.

### 4.11. Statistics

Statistical analysis and graph plotting were performed using GraphPad Prism software (version 8.0.1; GraphPad Software, San Diego, CA, USA). To identify significant differences between the different conditions, normality and homoscedasticity were confirmed, followed by One-way or Two-way ANOVA and a *post hoc* multiple comparison Dunnett’s test. Statistically significant differences were considered when the *p*-value was ≤0.05.

## 5. Conclusions

This study developed and validated a stepwise screening pipeline for selecting probiotic *Bacillus* strains that can protect host cells from *L. monocytogenes*. A total of 26 isolates were evaluated for antimicrobial activity, gastrointestinal resilience, and epithelial adhesion. A fluorescence-based infection assay using mCherry-HCT 116 cells was implemented as a predictive and high-throughput functional screen, identifying eight strains with intracellular protective activity. Among these, *B. subtilis* CECT 8266 showed the most potent effects, fully blocking *L. monocytogenes* replication in host cells, significantly reducing splenic bacterial burden in a murine model, and exhibiting strong host cell adhesion and bile/pH tolerance. Genomic analysis confirmed the absence of functional laterally acquired antibiotic resistance genes and revealed biosynthetic potential for multiple known anti-*Listeria* bacteriocins. These results validate the use of host cell-based fluorescence assays as efficient predictors of in vivo efficacy, supporting the translational potential of *B. subtilis* CECT 8266 as a safe and promising candidate for food biocontrol or therapeutic applications.

## Figures and Tables

**Figure 1 antibiotics-14-00765-f001:**
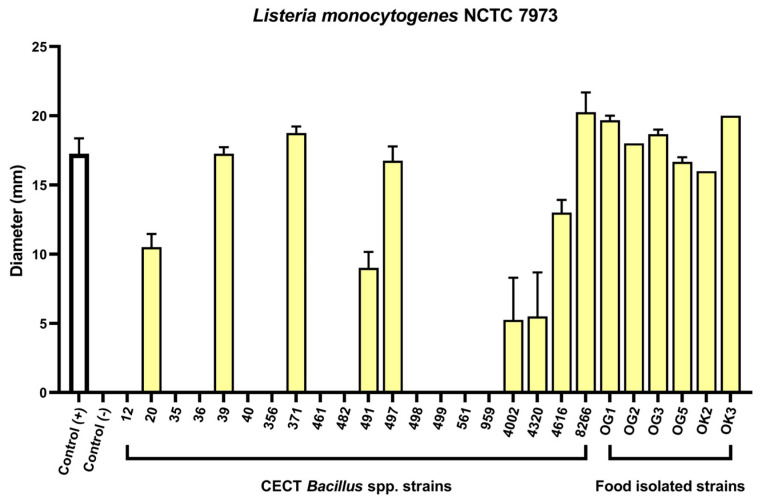
Bioassay results of cell-free supernatants from all *Bacillus* spp. strains tested against *L. monocytogenes* NCTC 7973. Positive and negative controls were TSB medium with 50 µg/mL kanamycin (+) or without kanamycin (−).

**Figure 2 antibiotics-14-00765-f002:**
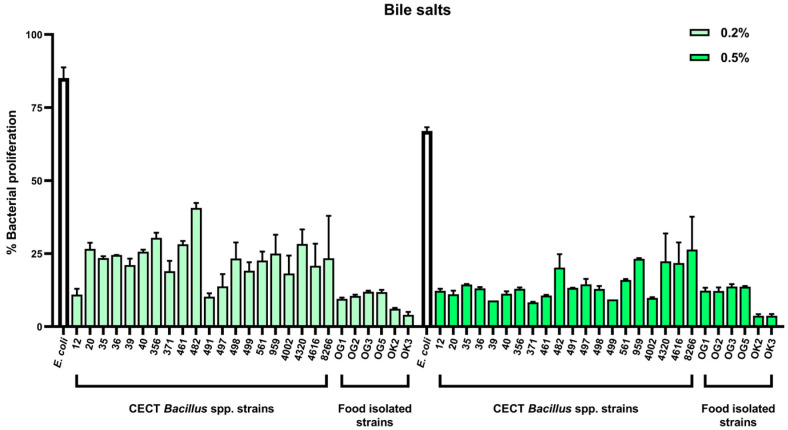
Proliferation levels of all *Bacillus* spp. strains cultured in TSB supplemented with varying concentrations of bile salts. Growth is depicted at different bile salt concentrations to evaluate the tolerance of each strain compared to a control grown in TSB without bile salts. *E. coli* ATCC 25922 was used as a reference strain.

**Figure 3 antibiotics-14-00765-f003:**
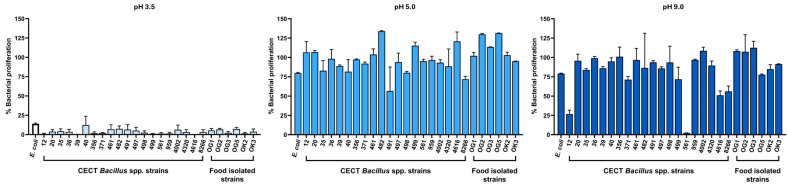
Proliferation levels of all *Bacillus* spp. strains cultured in TSB adjusted to different pH levels. Growth was measured under various pH conditions to assess the tolerance of each strain relative to growth at pH 7.0. *E. coli* ATCC 25922 was used as a reference strain.

**Figure 4 antibiotics-14-00765-f004:**
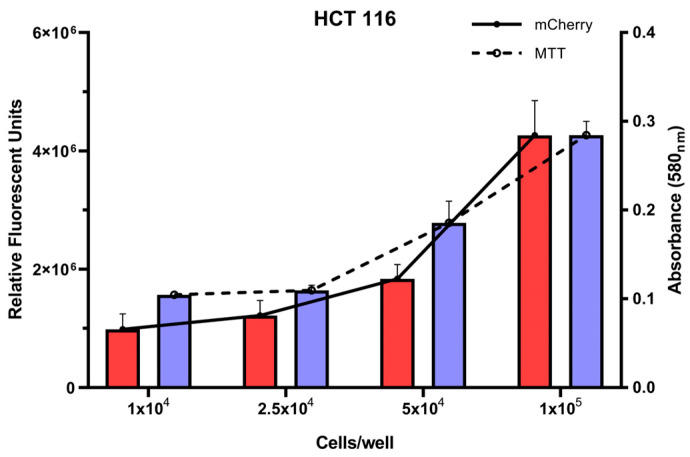
Comparison of the fluorescence signal produced by mCherry-expressing HCT 116 cells with MTT assay results, in relation to increasing cell numbers. The graphs illustrate the correlation between fluorescence intensity and MTT levels as cell density increases.

**Figure 5 antibiotics-14-00765-f005:**
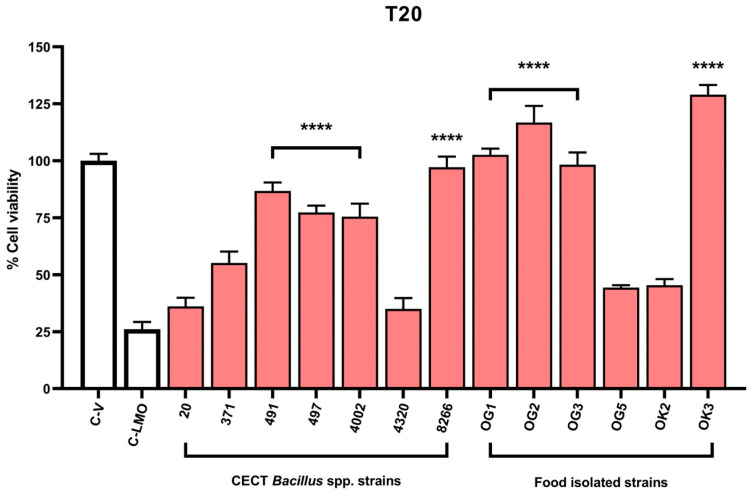
Host cell survival of HCT 116 cells treated with different *Bacillus* spp. strains following infection with *L. monocytogenes* NCTC 7973. The figure illustrates the protective effect of *Bacillus* spp. on host cell viability at 20-h post-infection (T20). Data were normalized to the negative control (C-V), consisting of uninfected and untreated cells. All comparisons were made relative to the positive control (C-LMO), consisting of infected but untreated cells. Data represent the result of at least three independent experiments ± SEM. Statistical significance between groups was analysed by *t*-test; *p*-value < 0.0001 ****.

**Figure 6 antibiotics-14-00765-f006:**
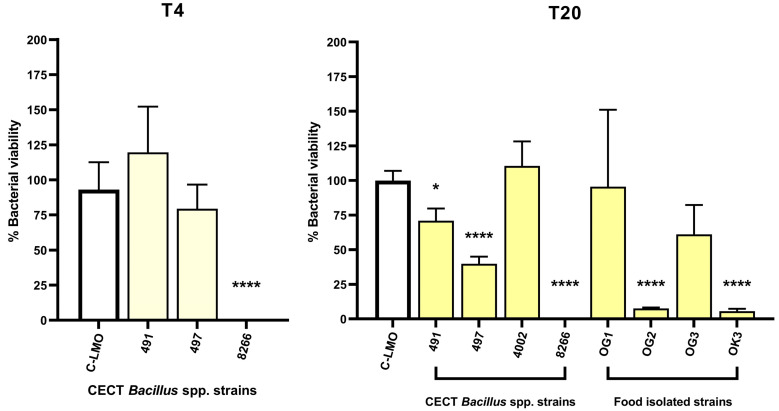
*L. monocytogenes* NCTC 7973 survival during HCT 116 infection in cells treated with different *Bacillus* spp. strains at 4- or 20-h post-infection (T4/T20). The figure highlights the impact of *B. subtilis* CECT 8266 and OK3 treatment on the survival of *L. monocytogenes* in the infected host cells. All comparisons were performed relative to the positive control, which consisted of infected and untreated cells (C-LMO). Data represent the result of at least three independent experiments ± SEM. Statistical significance between groups was analysed by *t*-test; *p*-value < 0.05 *; *p*-value < 0.0001 ****.

**Figure 7 antibiotics-14-00765-f007:**
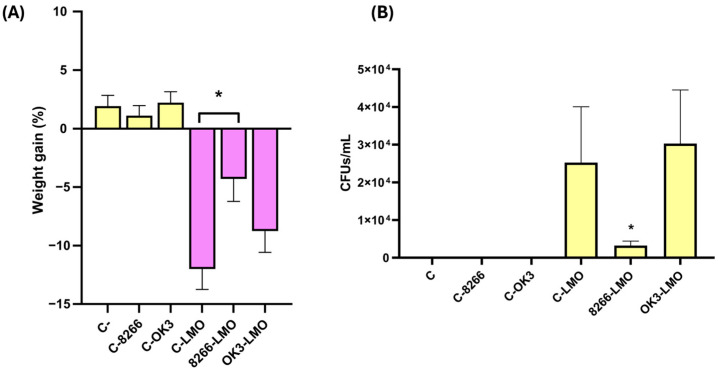
In vivo challenge results in BALB/c mice fed with *B. subtilis* CECT 8266 or OK3 strains, followed by *L. monocytogenes* NCTC 7973 infection. (**A**) Weight gain was assessed from day 11 to day 13 post-infection. (**B**) Bacterial load of *L. monocytogenes* in the spleens of infected mice on day 13 post-infection. No significant impact on body weight was observed in non-infected mice treated with either *B. subtilis* strain, indicating their safety. Infected mice experienced significant weight loss compared to non-infected controls ((**A**); purple bars). Only *B. subtilis* CECT 8266 significantly mitigated weight loss ((**A**); 8266-LMO) and reduced pathogen load in the spleens of infected mice ((**B**); 8266-LMO). All comparisons were performed relative to the positive control of infection, which consisted of infected and untreated mice (C-LMO). Statistical significance between groups was analysed by *t*-test; *p*-value < 0.05 *.

**Figure 8 antibiotics-14-00765-f008:**
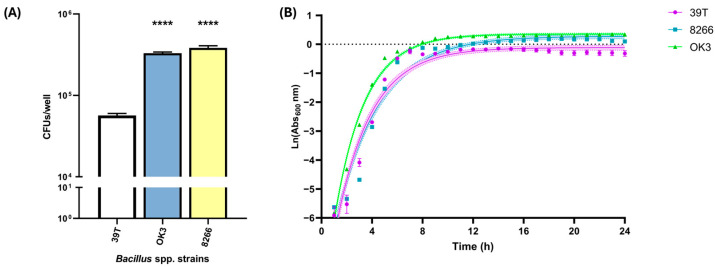
(**A**) Adhesion of three *Bacillus* spp. strains to HCT 116 enterocytes. *B. subtilis* OK3 and CECT 8266 exhibited higher adhesion compared to the type strain *B. subtilis* 39T. (**B**) Growth kinetics of three *Bacillus* spp. strains monitored by measuring absorbance at 600 nm every hour over 24 h. All comparisons were performed relative to the *B. subtilis* 39T. Data represent the result of three independent experiments ± SEM. Statistical significance between groups was analysed by *t*-test; *p*-value < 0.0001 ****.

**Figure 9 antibiotics-14-00765-f009:**
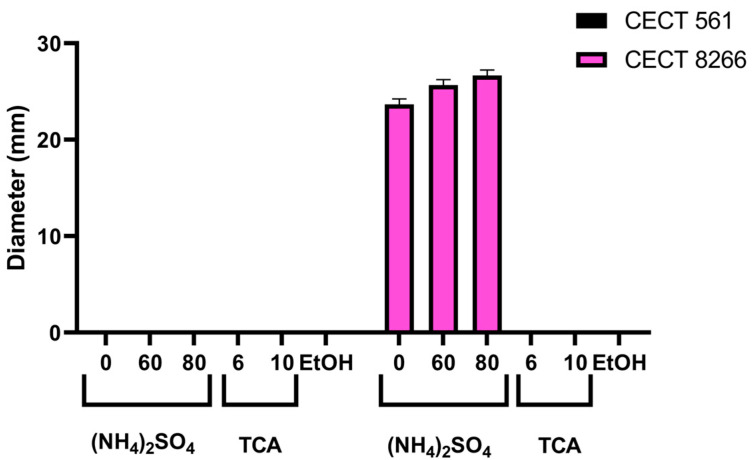
Bacteriocin activity of protein fractions precipitated from culture supernatants of two *Bacillus* spp. strains (CECT 561 and 8266) using ammonium sulphate, ethanol, or trichloroacetic acid (TCA). Inhibition zones indicate antimicrobial activity against *L. monocytogenes* NCTC 7973. Left and right panels correspond to different *Bacillus* isolates (CECT 561 and 8266, respectively).

## Data Availability

The complete genome sequence of *Bacillus subtilis* CECT 8266 has been deposited in the NCBI database under BioProject accession number PRJNA1167475, BioSample accession number SAMN44006356, and GenBank accession number CP170582.

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
