# Peer review of "Streamlining Bacillus Strain Selection Against Listeria monocytogenes Using a Fluorescence-Based Infection Assay Integrated into a Multi-Tiered Validation Pipeline"

_antibiotics, 2025, doi:10.3390/antibiotics14080765_

Round 1
Reviewer 1 Report
Comments and Suggestions for Authors
Abstract
Lines 25-28: did the authors use only one strain of L.monocytogenes? please indicate the source of isolation…
Introduction
I would suggest authors to expand the abstract by mentioning the recent developments of this method that is used in the food science studies.
Lines 102-103: please provide the definition of “small intestinal conditions” in details.
Lines 193-140: what is the correlation coefficient? Please indicate..
Lines 223-224: how did the authors calculate the growth rate? Fig.8B only shows the absorbance values of the selected bacteria.
Lines 237-260: section 2.6 provides the methods of bacteriocin recovery. Please revise this section and mention the key finding regarding to bacteriocin recovery.
Lines 292-293: please add references to support this statement.
I suggest authors to expand the discussion part of the article. Please discuss the key finding of this study with the existing literature. Possibly, indicate the limitations (if there are any..) and improvements compared to existing methods. As it stands, the discussion part seems a rather feeble and needs to be improved.
Lines 380-386: please indicate the source of L.monocytogenes..
Conclusion is one of the most important parts of an article. In this context, please mention the key findings of your study in the conclusion instead of general statements. Conclusion part needs to be improved and revised.
Author Response
Abstract
Lines 25-28: did the authors use only one strain of L.monocytogenes? please indicate the source of isolation…
In this case, the source of isolation is the British National Collection of Type Cultures (NCTC), and we have added this information in line 26: L. monocytogenes NCTC 7973
Introduction
I would suggest authors to expand the abstract by mentioning the recent developments of this method that is used in the food science studies.
We have expanded the introduction by adding the following sentence in lines 65-67:
Recent studies have explored host cell-based screening methods to evaluate the efficacy of probiotic and biocontrol candidates against foodborne pathogens in vitro.
This sentence is based on the following references:
Rani, R. P., et al. (2016). Front. Microbiol. [https://doi.org/10.3389/fmicb.2016.01910].
Drolia, R. et al. (2020). Nat. Commun. 11, 6344. [https://doi.org/10.1038/s41467-020-20200-5]
Lines 102-103: please provide the definition of “small intestinal conditions” in details.
We have added the requested details in the current 101-106 lines:
We therefore assessed the capacity of all Bacillus spp. strains to grow in the presence of bile salts (up to 0.5%) and at acidic pH values (as low as pH 3.5) to simulate small intestinal conditions. In adults, bile salt concentrations in the duodenum can reach ~10 mM (≈0.46% w/v) postprandially, while pH increases rapidly from ~3 in the stomach to ~7 in the upper small intestine, reflecting a major physicochemical transition relevant for probiotic survival [29].
Lines 193-140: what is the correlation coefficient? Please indicate..
Thanks, the correlation coefficient has been calculated and it is equal to 0.92
This information is included in the current lines 142-143:
A strong linear correlation was observed (Fig. 4; correlation coefficient = 0.92)
Lines 223-224: how did the authors calculate the growth rate? Fig.8B only shows the absorbance values of the selected bacteria.
We have analysed the slope of the growth curves in the current version, and this text was added to the results section (lines 230-240)
“B. subtilis CECT 39T exhibited the highest exponential growth rate (slope = 0.2736), followed by B. subtilis CECT 8266 (0.2574) and B. subtilis OK3 (0.1832). However, this growth rate did not correlate with final biomass accumulation, as B. subtilis OK3 reached the highest optical density, while B. subtilis CECT 39T reached the lowest. Despite these in vitro differences, only strain B. subtilis CECT 8266 conferred protection in the murine infection model. Neither the fastest-growing strain (B. subtilis CECT 39T) nor the strain reaching the highest OD (B. subtilis OK3) showed protective effects. These results indicate that growth kinetics alone do not predict in vivo efficacy, and suggest that strain-specific factors beyond replication rate or biomass accumulation likely contribute to the probiotic potential of B. subtilis CECT 8266.”
Lines 237-260: section 2.6 provides the methods of bacteriocin recovery. Please revise this section and mention the key finding regarding to bacteriocin recovery.
We have revised Section 2.6 to highlight the main findings regarding bacteriocin recovery explicitly (current lines 253-283). Specifically, we now clarify that ammonium sulphate precipitation successfully recovered active antimicrobial compounds from B. subtilis CECT 8266, while TCA and ethanol precipitation did not, and that no activity was detected in the negative control strain. This supports the presence of secreted, proteinaceous antimicrobials consistent with bacteriocins.
Lines 292-293: please add references to support this statement.
We have now added appropriate references to support this statement (line 307). Specifically, the revised text cites the EFSA Guidance on the Assessment of Bacterial Susceptibility to Antimicrobials of Human and Veterinary Importance (2012), which outlines the requirement for the absence of acquired antimicrobial resistance for strains to qualify under the Qualified Presumption of Safety (QPS) framework. We also cite Elshaghabee et al. (2017), which discusses the safety considerations for Bacillus strains used in probiotic applications, reinforcing the importance of ensuring the absence of resistance determinants.
I suggest authors to expand the discussion part of the article. Please discuss the key finding of this study with the existing literature. Possibly, indicate the limitations (if there are any..) and improvements compared to existing methods. As it stands, the discussion part seems a rather feeble and needs to be improved.
This comment partially contradicts a suggestion made by Reviewer 2. To address both reviewers' concerns, we have refrained from substantially expanding the Discussion section. Instead, we have streamlined the text by removing content that repeated information from the Results section, and we have focused on providing a deeper interpretation of the key findings. Additionally, we now highlight some limitations of the study, particularly those related to the in vivo infection model (lines 374-382) and the purification of antimicrobial compounds (lines 383-389).
Lines 380-386: please indicate the source of L.monocytogenes..
The origin of the Listeria monocytogenes strains is described in Supplementary Table S1. To improve clarity, we have now also added a dedicated paragraph in Section 4.1 of the Materials and Methods describing the strains used and their respective sources (lines 405-409).
Conclusion is one of the most important parts of an article. In this context, please mention the key findings of your study in the conclusion instead of general statements. Conclusion part needs to be improved and revised.
We have thoroughly revised the conclusion to include the key quantitative and qualitative findings of the study. The revised text now summarizes the number of Bacillus strains screened, the implementation of a fluorescence-based host cell assay, the identification of B. subtilis CECT 8266 as the most effective strain, and the supporting in vivo and genomic data that validate its potential as a safe and functional biocontrol agent (lines 644-657).
Reviewer 2 Report
Comments and Suggestions for Authors
Review of the manuscript antibiotics-3743415 entitled “Streamlining Bacillus strain selection against Listeria monocytogenes using a fluorescence-based infection assay integrated into a multi-tiered validation pipeline”, submitted to the journal Antibiotics:
- The abstract includes the aim, methodology, results, and conclusions of the study; however, it lacks quantitative information that would allow the reader to assess the scale of the research. For instance, the number of strains tested at each stage (e.g., those selected for in vivo testing) is not provided. Vague expressions such as “multiple bacteriocin biosynthetic clusters” and “significantly reducing bacterial burden” should be replaced with precise numerical data.
- The manuscript does not clearly specify which primer sets were used for genetic analysis of all the strains (see lines 595–596). This information is essential for reproducibility.
- The in vivo experiments on mice do not mention mortality or survival rates. The authors focus solely on CFU counts and body weight, which provides an incomplete picture of the outcome.
- There is no information on the threshold CFU reduction considered biologically meaningful. Only graphs are presented, with no indication of what constitutes significant reduction from a biological standpoint.
- The discussion section is overly lengthy and, in several places, redundantly reiterates results already described. It should be substantially shortened and refocused on interpretation rather than repetition.
- The genomic data are insufficiently interpreted. Although the authors state that the detected clusters are “typical”, no analysis of gene expression (transcriptomic or proteomic) is presented to support this conclusion.
- The potential role of secondary metabolites other than bacteriocins is not discussed, despite their likely relevance to the observed antimicrobial activity.
- Figure 9B is difficult to interpret due to scale and resolution issues. The visual complexity impairs its informational value.
- Lines 31–32: The phrase “completely inhibiting intracellular replication” should be accompanied by specific CFU reduction values to allow verification and interpretation.
- Line 33: “no acquired antibiotic resistance genes” — it is necessary to specify which resistance genes were analyzed and whether they were detected by phenotypic or genomic methods.
- Line 75: The term “regulatory compatibility” is vague and should be replaced with more precise wording such as “safety and regulatory compliance”.
- Line 248: The claim of “irreversible denaturation” lacks a supporting reference. A citation should be provided or the statement rephrased to reflect the evidence.
- Line 294: “No resistance was observed” — this should be clarified with MIC values or at least the detection thresholds applied in the testing.
- Lines 361–362: The statement that B. subtilis CECT 8266 is “a robust candidate” is too strong, especially given that no clinical or advanced preclinical trials have been conducted. A more cautious formulation is recommended.
- Lines 370–371: The suggested application in “cleaning systems for food-processing environments” is not experimentally supported. The manuscript does not include any data on biofilm formation or survival on surfaces.
I recommend that the manuscript be revised thoroughly before further consideration for publication.
Author Response
Review of the manuscript antibiotics-3743415 entitled “Streamlining Bacillus strain selection against Listeria monocytogenes using a fluorescence-based infection assay integrated into a multi-tiered validation pipeline”, submitted to the journal Antibiotics:
1. The abstract includes the aim, methodology, results, and conclusions of the study; however, it lacks quantitative information that would allow the reader to assess the scale of the research. For instance, the number of strains tested at each stage (e.g., those selected for in vivo testing) is not provided. Vague expressions such as “multiple bacteriocin biosynthetic clusters” and “significantly reducing bacterial burden” should be replaced with precise numerical data.
We have revised the abstract to include key quantitative data, such as the total number of Bacillus isolates screened (26), the number of strains that showed protective activity in the fluorescence assay (8), and the number selected for in vivo validation (2). We also replaced vague expressions with more precise descriptions, specifying that B. subtilis CECT 8266 harbours at least eight bacteriocin biosynthetic clusters and reduced splenic L. monocytogenes burden by 6-fold in treated mice. These changes enhance clarity and facilitate a more accurate assessment of the study’s scale and outcomes (lines 20-38).
2. The manuscript does not clearly specify which primer sets were used for genetic analysis of all the strains (see lines 595–596). This information is essential for reproducibility.
We have now explicitly indicated in the Materials and Methods (Section 4.10, lines 616–621) that all Bacillus strains selected from fermented foods were taxonomically characterized using 16S rDNA and tuf gene sequencing. The primer sets used for these analyses were 8FPL/806R for 16S rDNA and tuf2-F/tuf2-R for tuf, as previously described.
3. The in vivo experiments on mice do not mention mortality or survival rates. The authors focus solely on CFU counts and body weight, which provides an incomplete picture of the outcome.
We have now clarified in the revised manuscript that survival was monitored throughout the in vivo experiment. Only one animal died in the infected, untreated control group during the 48-hour post-infection period. All remaining mice in all groups—including those treated with Bacillus strains—survived until the endpoint and were euthanised for analysis. This information has now been included in Section 2.4 (lines 197–200).
4. There is no information on the threshold CFU reduction considered biologically meaningful. Only graphs are presented, with no indication of what constitutes significant reduction from a biological standpoint.
We thank the reviewer for this important point. Mice treated with B. subtilis CECT 8266 exhibited a ~6-fold reduction in splenic L. monocytogenes counts compared to the infected, untreated control group. This is a statistically significant change, and we consider it biologically meaningful as it was accompanied by complete survival and significantly less weight loss, whereas one animal in the control group died. This combination of microbiological and clinical outcomes supports the conclusion that B. subtilis CECT 8266 confers protection in vivo. We have revised the Results and Discussion sections accordingly to clarify this point (Lines 213-221 & 374-382).
5. The discussion section is overly lengthy and, in several places, redundantly reiterates results already described. It should be substantially shortened and refocused on interpretation rather than repetition.
Both reviewers requested revisions to the Discussion section, and we have therefore undertaken a substantial revision. The section has been refocused to emphasise interpretation of the results, as recommended (lines 334-400).
6. The genomic data are insufficiently interpreted. Although the authors state that the detected clusters are “typical”, no analysis of gene expression (transcriptomic or proteomic) is presented to support this conclusion.
We agree with the reviewer that genomic detection of biosynthetic gene clusters does not confirm their expression or functional activity. In our study, the annotation of NRPS and bacteriocin-related clusters was performed using BAGEL5 and antiSMASH, and we referred to them as “typical” in the sense that their structure and organisation closely resemble well-characterised clusters from Bacillus species. However, we acknowledge that no transcriptomic or proteomic data are currently available to confirm their expression under the conditions tested. We have revised the manuscript to reflect this limitation in the Discussion (lines 383-389).
7. The potential role of secondary metabolites other than bacteriocins is not discussed, despite their likely relevance to the observed antimicrobial activity.
We thank the reviewer for this valuable comment. In our study, antimicrobial activity was recovered from the ammonium sulfate precipitate, a fraction expected to be enriched in small proteins and peptides, but largely devoid of small secondary metabolites. Consequently, our discussion focused on proteinaceous compounds, particularly bacteriocins. However, we acknowledge that Bacillus strains are known to produce a broad spectrum of bioactive secondary metabolites, and their potential contribution to the observed activity cannot be excluded without further purification and analytical characterization. We have now addressed this methodological limitation in the revised Discussion section (lines 386–389), as part of our response to the previous comment.
8. Figure 9B is difficult to interpret due to scale and resolution issues. The visual complexity impairs its informational value.
We appreciate the reviewer’s observation. Figure 9B indeed contains detailed structural information that is difficult to visualize at the current resolution and scale. To improve clarity and maintain access to the full data, we have moved this panel to the Supplementary Material (Figure S5). We believe this preserves its value without compromising the readability of the main manuscript.
9. Lines 31–32: The phrase “completely inhibiting intracellular replication” should be accompanied by specific CFU reduction values to allow verification and interpretation.
The original sentence has been revised to include quantitative information. In our intracellular infection assay using HCT116 cells, treatment with B. subtilis CECT 8266 reduced intracellular Listeria monocytogenes burden from 100% (infected, untreated control) to undetectable levels in all replicates, representing a complete inhibition of intracellular replication under the conditions tested. This has now been clarified in the revised version of the Abstract (Lines 30-31).
10. Line 33: “no acquired antibiotic resistance genes” — it is necessary to specify which resistance genes were analyzed and whether they were detected by phenotypic or genomic methods.
We have now clarified that antibiotic resistance genes were assessed using both Alien_Hunter and the Comprehensive Antibiotic Resistance Database (CARD). Alien_Hunter was applied to identify genomic islands potentially acquired through horizontal gene transfer, while CARD was used to detect known acquired resistance determinants. Although some putative resistance genes were identified within genomic islands, these appear to be silent, as they do not confer phenotypic resistance to vancomycin, streptomycin, or kanamycin. We have updated the Abstract accordingly (line 33), and discussed this point in lines 390-395.
11. Line 75: The term “regulatory compatibility” is vague and should be replaced with more precise wording such as “safety and regulatory compliance”.
Agreed, the term “regulatory compatibility” has been replaced with “safety and regulatory compliance” to improve clarity and align with standard terminology used in risk assessment and probiotic evaluation (lines 76-78).
12. Line 248: The claim of “irreversible denaturation” lacks a supporting reference. A citation should be provided or the statement rephrased to reflect the evidence.
We agree that the original statement implied a mechanism (irreversible denaturation) that was not directly demonstrated in our study and was insufficiently supported by the provided references. Therefore, we have rephrased the sentence to clearly reflect our experimental results without speculating on the mechanism:
"Ammonium sulphate precipitation was selected as the initial step for recovering bacteriocin activity due to its mild, tunable conditions, which allowed for the successful recovery of bioactive peptides from B. subtilis CECT 8266. In contrast, other methods, such as tri-chloroacetic acid or ethanol precipitation, did not yield active fractions under our experimental conditions [38,44,45]."
This revision has been included in lines 261–265 of the revised manuscript.
13. Line 294: “No resistance was observed” — this should be clarified with MIC values or at least the detection thresholds applied in the testing.
We have provided the minimum inhibitory concentration (MIC) values determined for B. subtilis CECT 8266 against a panel of EFSA-recommended antibiotics. These values, along with the EFSA-defined resistance thresholds, are detailed in Table S3. Specifically, MIC values obtained for B. subtilis CECT 8266 were below EFSA's defined resistance cut-offs for all tested antibiotics, confirming the absence of phenotypic antibiotic resistance in this strain.
14. Lines 361–362: The statement that B. subtilis CECT 8266 is “a robust candidate” is too strong, especially given that no clinical or advanced preclinical trials have been conducted. A more cautious formulation is recommended.
We have modified this sentence across the manuscript to propose that this is a promising candidate (e.g. lines 37 and 652).
15. Lines 370–371: The suggested application in “cleaning systems for food-processing environments” is not experimentally supported. The manuscript does not include any data on biofilm formation or survival on surfaces.
We acknowledge that the manuscript does not currently include experimental evidence on biofilm formation or survival of B. subtilis CECT 8266 on food-processing surfaces. Accordingly, we have removed that sentence.
I recommend that the manuscript be revised thoroughly before further consideration for publication.
We sincerely thank the reviewers for their detailed and thoughtful review. The manuscript has undergone a comprehensive revision, thoroughly addressing each specific point raised by both reviewers. We believe these revisions have substantially improved the manuscript's clarity, accuracy, and overall rigour. All modifications have been indicated in the revised manuscript.
Round 2
Reviewer 1 Report
Comments and Suggestions for Authors
The authors state that they analyze slope of the curves in the current version. However, they did not define which function they used to fit the growth results in the materials and methods section. In this context, the authors should provide the function they used to calculate the growth rates of the microorganisms. In addition to this, the authors were also advised that the y-axis should be changed in Fig. 8B from OD to growth rate and the final biomass should be mentioned in the text in terms of OD.
Author Response
Reviewer comment:
The authors state that they analyze the slope of the curves in the current version. However, they did not define which function they used to fit the growth results in the Materials and Methods section. In this context, the authors should provide the function they used to calculate the growth rates of the microorganisms. In addition to this, the authors were also advised that the y-axis should be changed in Fig. 8B from OD to growth rate and the final biomass should be mentioned in the text in terms of OD.
Response:
In the revised version of the manuscript, we have reanalyzed the bacterial growth kinetics using a semi-logarithmic approach, plotting ln(Abs₆₀₀) versus time during the exponential growth phase. We applied linear regression to calculate the specific growth rate (μ, h⁻¹) for each strain and report these values with three decimal places (lines 229–244, Figure 8B).
Additionally, we have replaced the term Optical Density with Absorbance throughout the text, as the data were obtained directly as Abs₆₀₀ measurements. The Materials and Methods section has been updated to include a detailed explanation of the regression method used, the intervals selected for each strain, and the regression equation (lines 433–441).
Reviewer 2 Report
Comments and Suggestions for Authors
The authors have adequately addressed all the reviewers’ comments and suggestions. The revised version is clear, well-structured, and scientifically sound. I recommend the manuscript for publication in its current form.
Author Response
Thank you again for all your work on the revision of this manuscript.